# Algorithmically Reconstructed Molecular Pathways as the New Generation of Prognostic Molecular Biomarkers in Human Solid Cancers

**DOI:** 10.3390/proteomes11030026

**Published:** 2023-08-25

**Authors:** Marianna Zolotovskaia, Maks Kovalenko, Polina Pugacheva, Victor Tkachev, Alexander Simonov, Maxim Sorokin, Alexander Seryakov, Andrew Garazha, Nurshat Gaifullin, Marina Sekacheva, Galina Zakharova, Anton A. Buzdin

**Affiliations:** 1Laboratory for Translational Genomic Bioinformatics, Moscow Institute of Physics and Technology (State University), 141701 Dolgoprudny, Russia; 2Omicsway Corp., Walnut, CA 91789, USA; 3Laboratory of Clinical and Genomic Bioinformatics, I.M. Sechenov First Moscow State Medical University, 119048 Moscow, Russia; 4PathoBiology Group, European Organization for Research and Treatment of Cancer (EORTC), 1200 Brussels, Belgium; 5Medical Holding SM-Clinic, 105120 Moscow, Russia; 6Department of Pathology, Faculty of Medicine, Lomonosov Moscow State University, 119991 Moscow, Russia; 7World-Class Research Center “Digital Biodesign and Personalized Healthcare”, Sechenov First Moscow State Medical University, 119048 Moscow, Russia; 8Laboratory of Systems Biology, Shemyakin-Ovchinnikov Institute of Bioorganic Chemistry, 117997 Moscow, Russia

**Keywords:** cancer, gene expression, molecular pathway, human interactome, prognostic biomarker, survival biomarker, RNA sequencing, proteomic data

## Abstract

Individual gene expression and molecular pathway activation profiles were shown to be effective biomarkers in many cancers. Here, we used the human interactome model to algorithmically build 7470 molecular pathways centered around individual gene products. We assessed their associations with tumor type and survival in comparison with the previous generation of molecular pathway biomarkers (3022 “classical” pathways) and with the RNA transcripts or proteomic profiles of individual genes, for 8141 and 1117 samples, respectively. For all analytes in RNA and proteomic data, respectively, we found a total of 7441 and 7343 potential biomarker associations for gene-centric pathways, 3020 and 2950 for classical pathways, and 24,349 and 6742 for individual genes. Overall, the percentage of RNA biomarkers was statistically significantly higher for both types of pathways than for individual genes (*p* < 0.05). In turn, both types of pathways showed comparable performance. The percentage of cancer-type-specific biomarkers was comparable between proteomic and transcriptomic levels, but the proportion of survival biomarkers was dramatically lower for proteomic data. Thus, we conclude that pathway activation level is the advanced type of biomarker for RNA and proteomic data, and momentary algorithmic computer building of pathways is a new credible alternative to time-consuming hypothesis-driven manual pathway curation and reconstruction.

## 1. Introduction

Variation of gene expression among individual tumors enables the personalization of many diagnostic and treatment options [1]. Indeed, multiple gene expression biomarkers were proposed for the prediction of patient survival and drug response (e.g., [1,2,3,4,5]). Several transcriptomic biomarkers were approved for clinical use, e.g., gene expression signatures predicting recurrence and prognosis in breast and thyroid cancers [6,7,8].

However, gene products do not act alone, but rather as the components of complex molecular networks executing specific functions in cell molecular physiology. Thus, cancer-specific alterations of gene expression inevitably lead to dysregulation of multiple molecular pathways [9]. This makes it possible to create the next generation of signatures based on molecular pathway activities and investigate their association with various characteristics of cancers, e.g., tumor grade, invasiveness and histological type, patient survival, and response to therapy [10,11,12,13]. Pathways affected in a tumor can be identified using various statistical methods for both RNA and protein expression data [14,15,16,17]. Alternatively, gene ontology (GO) analysis can identify molecular processes enriched by the differentially expressed genes [18].

Many such approaches ignore pathway functional topology and fail to determine the up- or downregulated state of a pathway and the extent of its activation. Indeed, different components of a molecular pathway may have different functional roles (for example, increased expression of an inhibitory component would act in favor of pathway downregulation, and vice versa). Furthermore, pathways may include feedback loops and other complex interactions that have to be taken into consideration when quantitatively assessing pathway alterations in cancer [19,20].

In order to translate expression data into quantitative measures of pathway deregulation while considering pathway architecture, a measure termed Pathway Activation Level (PAL) was introduced [14,21,22]. For a given molecular pathway, PAL is calculated as a weighted sum of logarithms of case-to-normal ratios for the expression levels of all genes involved in the pathway of interest, with weights ranging from –1 to 1 according to the activator/repressor role of the corresponding gene products.

When used to discriminate between nine human cancer types, PAL values showed better accuracy than expression levels of individual genes [23]. On its own, PAL was also found to be a good predictor of tissue type in bladder cancer [9] and of sensitivities to some cancer drugs [13,24,25,26,27,28]. Additionally, PALs demonstrated better stability against experimental noise and lower batch bias compared to single gene expression levels in both transcriptomic (microarray and RNAseq) and proteomic data [14,23,29]. These findings suggest the advantage of PAL values or other metrics of that kind as potential molecular biomarkers.

Furthermore, a computational recursive approach was proposed that can algorithmically annotate activator/repressor roles to all pathway nodes depending on the pathway molecular architecture and the nature (activation/inhibition/other) of each molecular interaction within the pathway [20]. This enables fast, uniform, and simultaneous annotation of thousands of molecular pathways [30]. In addition, this approach also excludes the operator error that is a probable event during manual annotation of pathways and interactomes due to their high complexities.

Recently we published an alternative concept of a molecular pathway that is built algorithmically as an interacting network around the central node—gene product of interest [31]. This approach is based on the whole-interactome model and is fully automatic. It has the advantage of reducing bias introduced during manual reconstruction as in the case of the “classical” pathways. In the manually reconstructed pathways, the gene contents are typically investigator hypothesis-driven with a strong bias toward well-known “topical” molecules. As a result, such featured molecules are overrepresented in classical pathways whereas the others can be ignored or overrepresented instead.

Thus, using the whole interactome model, we constructed a set of so-called gene-centric pathways: local subnetworks of interacting molecules consisting of a central gene (main node of the pathway) and other molecular components interacting with this gene product either directly or indirectly. The gene-centric pathway is characterized by a maximal number of molecular interactions starting at the central node and leading to every other node of the pathway (one or two interactions in the published reports) [31]. One such algorithmically constructed pathway centered at gene *FREM2* emerged as a promising predictor of tumor grade and survival in human gliomas, strongly exceeding the biomarker performance of the *FREM2* gene itself [31]. We then investigated a larger number of gene-centric pathways in human gliomas, where they demonstrated an overall superior diagnostic and prognostic performance compared to single gene expression levels [32].

However, the relative performance of the gene-centric pathways in comparison to classical pathways and single genes remained unexplored at the pan-cancer level. Here we used a human interactome model involving 7470 human gene products to algorithmically reconstruct molecular pathways termed gene-centric pathways, centered around each of these genes. We then assessed their general biomarker characteristics in comparison with the previous generation of molecular pathways (3022 “classical” pathways) and with the transcripts of 24,862 individual genes. To this end, we investigated potential biomarker associations with tumor type and overall and progression-free survival in 21 human cancer types using RNA sequencing and proteomic data for 8141 and 1117 samples, respectively. For all analytes in RNA and proteomic data, respectively, we found a total of 7441 and 7343 potential biomarker associations for gene-centric pathways, 3020 and 2950 for classical pathways, and 24,349 and 6742 for individual genes. Overall, the percentage of potential RNA biomarkers was statistically significantly higher for both types of pathways than for individual genes (*p* < 0.05). In turn, both types of pathways showed comparable performance. While the percentage of potential cancer type-specific biomarkers was comparable between proteomic and transcriptomic levels, the proportion of potential survival biomarkers was dramatically lower for the proteomic data: up to only 2.3% versus as much as 36.3% in transcriptomic data. Thus, we conclude that pathway activation level is the advanced type of cancer biomarker for RNA data, and momentary algorithmic computer building of pathways is a new credible alternative to time-consuming hypothesis-driven manual pathway reconstruction.

## 2. Materials and Methods

### 2.1. Interactome Model and Gene-Centric Molecular Pathways

A model of the human interactome was built using a set of pairwise molecular interactions extracted from the annotations of 51,672 human molecular pathways. For this, we used the biggest available collection of molecular interaction-validated molecular pathways previously described in [30]. The collection includes pathways from the following six databases: Reactome [33], NCI Pathway Interaction Database [34], Biocarta [35], HumanCyc [36], QIAGEN-Pathway-Central, and PathBank [37]. An overall interactome model was built as a directed graph with nodes representing gene products or metabolites and edges standing for direct molecular interactions. The model is publicly available following reference [31] for our previously published paper where we assessed the biomarker capacity of gene-centric pathways for human gliomas. The interactome includes known metabolic reactions and transport processes as well as protein–protein interactions. The first two types of processes are marked with specific assistive nodes. All interactions were categorized and labeled as “activation”, “inhibition”, or “other”. The “other” interaction type includes interactions that cannot be related to activation or inhibition. Examples of “other” interaction types are “SubPathwayInteraction”, “ComplexAssembly”, “Compound”, and “Indirect values”. In total, the model built contains 64,095 molecular participants including 7496 gene products and 361,654 pairwise interactions, schematized in Figure 1.

Specific gene-centric molecular pathways were constructed for all genes from the interactome graph with a distance of one interaction from the central node. If one of those nodes was a “process” node (biochemical reaction or transport process), then all participants of that process were also included in the respective gene-centric pathway (7496 gene-centric pathways primarily built).

The pathway annotation algorithm was applied to each gene-centric pathway as previously described in [20] to assign activator-repressor role (ARR) coefficients to each component of every pathway, which defines their pathway activator, inhibitor, or neutral roles. The pathways having all ARR coefficients equal to zeros (2 pathways), or having either zero ARRs or no available expression data for all nodes (24 pathways) were excluded from further analysis, thus giving the total number of 7470 gene-centric pathways for further analysis. The reconstructed gene-centric pathways are added to the public pathway databank OncoboxPD, available at open.oncobox.com (accessed on 20 May 2023).

### 2.2. Classical Molecular Pathways

In this study, previously annotated molecular pathways taken from the specific pathway databanks were referred to as classical molecular pathways. A total of 3022 such pathways were extracted from the OncoboxPD collection [30].

### 2.3. Gene Expression Data and Clinical Annotation

For transcriptomic data, we used clinically annotated solid cancer RNA sequencing gene expression data of solid tumors and corresponding normal tissues from The Cancer Genome Atlas (TCGA) project repository [38]. In total, 21 cancer types were selected for the analysis, each having at least 100 available primary solid tumor samples (Table 1). For proteomic data, we used available datasets from the Proteomic Data Commons (PDC) portal with profiles for both tumor and normal samples (Table 1). Another inclusion criterion for proteomic data was label-based quantitation (iTRAQ, TMT).

We excluded from the analysis TCGA-LGG (low-grade gliomas) and TCGA-GBM (glioblastomas multiforme) cancers because they were recently characterized in our previous communication [32]. RNA sequencing gene expression data (STAR-counts) and clinical annotations (XML and BCR Biotab files for TCGA and PDC patient annotations) were downloaded from the GDC Data Portal, release v32.0 (portal.gdc.cancer.gov, accessed on 1 April 2022). Proteomic gene expression profiles (iTRAQ and TMT log-transformed ratios) were extracted from PDC Data Portal (pdc.cancer.gov, accessed on 1 April 2022).

For consistency, some projects (marked with asterisks (*) in Table 1) were simplified by histological type: a dominant tumor type (including possible subtypes) was identified, and samples of other histological types were excluded from the analysis. For example, in the TCGA-PCPG (Pheochromocytoma and Paraganglioma) group, only pheochromocytoma samples were considered.

### 2.4. Calculation of Pathway Activation Levels

To assess the activation status of molecular pathways using gene expression data relative to controls, we calculated Pathway Activation Levels, PALs [21]. For pathway *p*, PAL is calculated as follows:(1)$${PAL}_{p}=\sum_{n}^{\;}\log\left(CNR_{n}\right)\;\cdot \;{PG}_{p,n}\;\cdot \;{ARR}_{p,n}/\sum_{n}^{\;}|{ARR}_{p,n}|,$$

The sum is calculated over all genes in the assay; *PG_p,n_* = 1 if gene *n* is involved in pathway *p* and 0 otherwise. *CNR_n_* (case-to-normal ratio of gene *n*) is the ratio of its expression level in the tumor sample under analysis to the geometric mean expression level in the control group. *ARR_p,n_* (activator-repressor role of gene *n*), is a coefficient that characterizes the effect of gene product *n* on the activity of pathway *p*. *ARR_p,n_* can take one of the following values: 1 if gene product *n* activates pathway *p*, −1 if it inhibits pathway *p*, 0.5 or −0.5 if it is rather an activator or repressor of pathway *p*, respectively, and 0 if the functional role of *n* in pathway *p* is unclear or inconsistent. For all PAL calculations of the classical pathways, we used our publicly accessible OncoboxPD online tool open.oncobox.com [30].

### 2.5. Statistical Analysis of Potential Cancer-Specific Biomarkers

To detect potential cancer type-specific biomarkers, all samples under analysis were merged and further spread into two groups: (*i*) cancer type of interest and (*ii*) the rest.

In order to identify potential tumor biomarkers, samples of cancerous tissues were compared to samples of normal tissue from the corresponding anatomical site.

For measuring pathway and gene activities, PAL values and gene expression levels were analyzed, respectively, by assessing the Area Under the ROC Curve (AUC) and Wilcoxon paired test *p*-value. The *p*-values were FDR adjusted according to the Benjamini–Hochberg method [39]. Pathways/genes were considered statistically significant potential biomarkers in case of an AUC exceeding 0.7 and adjusted *p*-value (q) less than 0.05. All genes and classical and gene-centric pathways were checked for being significant potential biomarkers for each cancer type. For proteomic data, we used ratios of tumor profiles to average normal profile on the gene level to exclude the impact of a reference sample from the specific proteomic project.

### 2.6. Statistical Analysis of Survival Characteristics

For each cancer type and each gene or pathway, patients were classified into two groups depending on whether their gene expression or pathway activation level was above or below the median level. Overall and progression-free survival values were then assessed using the Kaplan–Meier method, and the statistical significance of differences between the two groups was determined by the *p*-value of the log-rank test. In each cancer type and each category of potential biomarkers (genes and gene-centric and classical pathways), the *p*-values were FDR-adjusted using the Benjamini–Hochberg method. The differences in survival characteristics were assessed by calculating hazard ratio (HR) in a univariate Cox model. Items with the adjusted *p*-value (*q*) < 0.05 and the HR confidence interval excluding 1 were considered significant. Genes and gene-centric and classical pathways were then compared for the percentage of statistically significant differential items and the corresponding HRs.

### 2.7. Software

Gene expression data were normalized with DESeq2 [40], and PAL values were calculated for gene-centric pathways using R [41]. PALs for classical pathways were calculated using the python package oncoboxlib [20].

Statistical analysis was carried out using R [41] with the following packages: pROC [42], Stats, survival [43,44], survminer [45], and survcomp [46,47]. Results were visualized in Python 3 using seaborn [48] and matplotlib [49], as well as in R using ggplot2 [50].

## 3. Results

### 3.1. Assessment of Potential Cancer-Type Biomarkers

The expression level of every gene was screened for the potential of serving as the potential cancer-type-specific biomarker. Similarly, pathway activation levels (PALs) were interrogated for all classical and gene-centric molecular pathways (Appendix A). For the 21 cancer types under consideration (Table 1) using both RNAseq and proteomic data, we identified a number of statistically significant potential biomarkers (AUC > 0.7 and adjusted *p*-value (q) < 0.05), Table 2.

In addition to the absolute values of potential biomarkers identified, we also calculated the percentage of statistically significant potential biomarkers for genes and classical and gene-centric molecular pathways (Figure 2A). We observed a statistically higher proportion of high-quality potential biomarkers for the molecular pathways (both classical and gene-centric) than for the individual genes (*t*-test *p*-value 0.042 and 0.038, respectively). Interestingly, at the same time, we found no significant difference between the biomarker capacities of gene-centric and classical pathways (*p* = 0.96).

We then assessed proteomic data (Table 1) in a similar way and found potential cancer-type-specific biomarkers at the levels of protein expression and activation levels of classical and gene-centric molecular pathways (Table 3 and Figure 2B). For proteomic data, we found no statistically significant difference between proportions of potential biomarkers at the level of proteins and gene-centric and classical pathways (Figure 2B).

We also screened for biomarker capacity expression levels of individual genes that serve as the central nodes for the corresponding gene-centric pathways, and PALs of the corresponding gene-centric pathways. Again, we observed a significantly higher proportion of high-quality potential biomarkers among the gene-centric pathways for RNAseq data (*p* = 1.25 × 10^−5^) but not for the proteomic profiles (Figure 3).

To assess any potential technical bias in the proteomic data used in this study, we examined the clustering of samples in relation to tumor type, tandem mass tag (TMT10 and TMT11), and the model of the mass spectrometer used (Figure 4). We quantitatively evaluated the quality of hierarchical clustering using the Watermelon multisection method [51]. This method provides a WM metric that indicates the effectiveness of sample clustering into predetermined groups, where a higher WM score means better clustering. The WM scores were 0.94, 0.98, and 0.99 for clustering by tandem mass tag, tumor type, and mass spectrometer model, respectively. Thus, we observed the strongest clustering by the type of spectrophotometer, about the same for tumor type, and less significant clustering by the type of tandem mass tag. Consequently, in high-throughput combinatory proteomic studies, attention should be paid to possible batch effects mainly connected with the equipment used. We then compared the average percentage shares of potential biomarkers for the datasets produced by the Orbitrap Fusion Lumos, Q Exactive HF-X, and Q Exactive Plus mass spectrometers (Table 4). We found that approximately 50% of the tested individual proteins and pathways were significant potential proteomic biomarkers in the datasets produced by the Orbitrap Fusion Lumos spectrometer. In contrast, up to 29% of proteins and up to 39% of pathways were detected as significant potential biomarkers in the datasets produced by spectrometers of the Q Exactive series. However, it is important to note that different spectrometers were used to interrogate cohorts of different cancer types and that apparent differences in biomarker abundance can be also related to the biological nature of the samples under analysis.

### 3.2. Assessment of Potential Tumor Biomarkers

To detect potential tumor biomarkers, we compared the expression levels of each gene in the tumor with those in the normal tissue. Similarly, we evaluated all classical and gene-centric molecular pathways by analyzing pathway activation levels (PALs) (Appendix A). Using RNAseq data, we found statistically significant potential biomarkers for 16 out of 21 considered cancer types, where the AUC > 0.7 and the adjusted *p*-value (q) < 0.05 (Table 5).

A higher proportion of high-quality potential biomarkers was observed for the classical molecular pathways compared to the individual genes (*t*-test *p*-value = 0.037, Figure 5A and Table 5). A similar trend was observed when comparing the gene-centric molecular pathways and the individual genes (*t*-test *p*-value = 0.096). Furthermore, we found no significant difference between the biomarker capacities of gene-centric and classical pathways (*t*-test *p*-value *p* = 0.72).

Besides the number or percentage of potential biomarkers, we compared the AUC distribution for the pathways and genes that were tested. The medians of the distributions for potential pathway-based biomarkers were higher than those for potential single-gene biomarkers (Appendix A). We performed a similar analysis of adjusted *p*-values, and the median of distribution was highest for genes (Appendix A). These results confirm the trend observed in the analysis of the proportions of significant potential biomarkers.

Potential tumor biomarkers were similarly detected by analyzing protein expression and activation levels of classical and gene-centric molecular pathways (Table 6 and Figure 5B). Our proteomic data analysis revealed a statistically higher proportion of high-quality potential biomarkers for the molecular pathways (both classical and gene-centric) compared to individual genes (*t*-test *p*-value of 1.5 × 10^−4^ and 1.5 × 10^−5^, respectively). Similar to the results of RNAseq analysis, there is no significant difference (*p* = 0.38) in the biomarker proportions between gene-centric and classical pathways.

We screened the proportion of significant potential tumor biomarkers at the level of individual genes that serve as the central nodes for the corresponding gene-centric pathways. Additionally, we screened PALs of the corresponding gene-centric pathways. We found a significantly higher proportion of high-quality potential biomarkers among the gene-centric pathways for proteomic data (*p* = 1.3 × 10^−4^), but not for the RNAseq profiles (*p* = 0.4), (Figure 6).

### 3.3. Assessment of Potential Survival Biomarkers

We then compared the biomarker capacity of the same three types of transcriptional variables for predicting patient overall survival and progression-free survival. Interestingly, for all types of available molecular data (genes and gene-centric and classical pathways), we identified statistically significant potential survival biomarkers for only 13 out of 21 cancer types. For both overall and progression-free survival, the biggest proportion of potential prognostic biomarkers of all three types was found in papillary and clear cell renal cell carcinomas, and in hepatocellular carcinoma. Moreover, a large number of potential biomarkers were detected for overall survival in head and neck squamous cell carcinoma and for progression-free survival in prostate adenocarcinoma (Figure 7 and Appendix A).

In order to estimate the likelihood of each category to provide a potential prognostic biomarker, we calculated the percentages of all genes and gene-centric and classical pathways that were linked with survival in different cancers and compared the results. For example, in clear cell renal cell carcinoma, as many as 7998 genes and 2110 gene-centric pathways were identified as potential prognostic biomarkers for overall survival, thus constituting 32.1% and 28.2% of all genes and gene-centric pathways, respectively; shown for all cancers in Figure 7A and Figure 8.

Of note, proteomic data analysis showed one or two orders of magnitude lower numbers of significant potential biomarkers than RNAseq data. We found that only pancreatic cancer had potential survival biomarkers at the level of single protein levels, and activation levels of gene-centric and classical molecular pathways (Figure 7B and Figure 8B). Breast, lung, and endometrial cancers each had small numbers of individual protein potential survival biomarkers.

Similar to the case of cancer type-specific biomarkers, gene-centric pathways and classical pathways demonstrated an advantage over individual genes for both types of survival biomarkers at the transcriptome level (Figure 8). Thus, for overall survival, the highest number of biomarkers is observed in two, five, and three cancer types for individual genes and gene-centric and classical pathways, respectively. For progression-free survival, the advantage has been shown in four, three, and five cancer types for individual genes and gene-centric and classical pathways, respectively.

At proteome level data, potential single protein biomarkers were found in four cancer types. Potential pathway-based biomarkers were found only for pancreatic cancer, with the highest percentage share of significant biomarkers among gene-centric pathways.

### 3.4. Prognostic Performance of Hazard Ratio for Overall Survival and Progression-Free Survival

Alternatively, the prognostic robustness of potential biomarkers was assessed at the level of hazard ratios (HRs) for overall survival and progression-free survival of cancer patients (Figure 8 and Figure 9). We compared distributions of HRs for cancer types with all three types of potential biomarkers: clear renal cell carcinoma, papillary renal cell carcinoma, hepatocellular carcinoma, head neck squamous cell carcinoma (overall survival only), and prostate adenocarcinoma (progression-free survival only). A majority of potential biomarkers for these cancers (except for clear renal cell carcinoma) are associated with poor prognosis independent of biomarker type (Figure 8A,B and Figure 9A,C). To assess if biomarker types for the same cancer differ by value of predicted risk, we compared absolute values of log-transformed HRs by the Wilcoxon test (Appendix A). We observed statistically significant differences between different biomarker types in most of the pairwise comparisons made; however, the magnitudes of these differences are low (0.001 to 0.17 between medians of HR distributions, Appendix A).

Additionally, we compared HR distributions for gene-centric pathways and their corresponding central genes (Figure 8A,B and Figure 9B,D). We identified significant differences only for progression-free survival in papillary renal cell carcinoma (Appendix A and Figure 9C,D) and for both types of survival data in hepatocellular carcinoma (Appendix A and Figure 10C,D). Potential individual gene biomarkers tended to have greater absolute values of log-transformed HRs than the corresponding gene-centric pathways for progression-free survival in papillary renal cell carcinoma and for overall survival in hepatocellular carcinoma. In contrast, gene-centric pathways were leading for progression-free survival in hepatocellular carcinoma (Appendix A). Nevertheless, despite the differences being statistically significant, their absolute values were relatively small.

Due to the small number of potential proteomic biomarkers, the same comparison could not be carried out at the level of proteomic data.

## 4. Discussion

We performed here the first pan-cancer screening including gene expression data for 21 human cancer types to compare the biomarker performance of manually and algorithmically reconstructed molecular pathways, and of individual genes. We found statistically significant cancer-type potential biomarkers in each cancer type under analysis, both among genes and gene-centric, and classical molecular pathways. The percentage of cancer-type biomarkers was significantly higher in both types of pathways (both gene-centric and classical) than among individual genes. The cancer-type-specific biomarkers may be important for a better understanding of tissue-specific aspects of carcinogenesis. In addition, we screened for potential biomarkers between tumors and normal tissues and observed the same trend that pathway-based potential biomarkers outperform single genes or proteins.

In 13 cancer types, we also identified putative prognostic biomarkers of all three types (genes and gene-centric and classical pathways). For overall survival, gene-centric pathways and classical pathways showed a higher percentage of significant potential biomarkers than individual genes in five and three cancer types, respectively, whereas potential gene biomarkers prevailed in two cancer types. For progression-free survival, the advantage has been shown, respectively, in four, three, and five cancer types for individual genes and gene-centric and classical pathways. Thus, we conclude that a pathway-based approach can result in enriched sets of potential biomarkers predicting survival than individual genes.

In terms of magnitudes of HRs associated with significant potential survival biomarkers, there were statistically significant yet relatively small differences between the above three biomarker types and no overall trend of an advantage of the certain biomarker type in all cancers.

Many previous studies attempted to link the activities of genes and their interacting networks with clinical outcomes [52,53,54,55,56,57]. In most of them, an overall analytic pipeline included assessment of differential gene expression and building co-expression networks, e.g., using Ingenuity Pathways Analysis [52] or by identification of fully connected gene sets enriched for certain functions, e.g., using the CytoScape ExpressionCorrelation tool [53]. Alternatively, genes could be grouped using weighted correlation network analysis (WGCNA) [58], e.g., for studying survival biomarkers in lung adenocarcinoma, in colon and renal cancers [54,55,56]. Protein–protein interactions from the STRING database (http://string-db.org/, accessed on 20 May 2023) were also used to supplement WGCNA for a more accurate prediction of patient survival in bladder cancer [57]. We tried to compare these approaches with the current study findings in terms of input data and output results in Table 7.

Thus, in this study, we considered not only the proximity of genes within topological interaction networks but also their functional roles. Unlike in the previous research, in addition to well-known classical pathways from popular databases, we also generated and in-depth analyzed algorithmically constructed gene-centric pathways.

Overall, the algorithmic approach was shown to be a robust method of obtaining new molecular pathways. The algorithm selected highly connected gene-centric subnetworks in the human interactome, and the molecular pathways obtained in such a way have demonstrated biomarker values comparable with pathways manually constructed by expert curation.

It is now widely accepted that a combination of biomarkers, such as gene signatures or pathways, is more robust and performs better than using individual genes or proteins. Our results confirm this trend. However, the number of algorithmically constructed pathways was about two times higher than for the source classical pathways. The ultra-fast speed and efficiency of this approach, therefore, make it a useful solution for hypothesis-free algorithmic annotation of the whole connectomes.

In the domain of tumor-type biomarkers, many studies rely on a deep learning approach [59,60,61], including convolutional neural networks [62,63]. However, to our knowledge, the only type of input data in such models was gene expression, and the nature of functional interactions within groups of genes generated was not considered. We speculate here that applying our gene-centric pathway approach, based on the whole-interactome model, to such deep learning settings, can further increase the biomarker capacity of both methods.

Besides gene expression values, we analyzed the biomarker capacity of proteins profiled using two labels (TMT10 and TMT11) and three models of mass spectrometers (Orbitrap Fusion Lumos, Q Exactive Plus, and Orbitrap Fusion Lumos). TMT11 and TMT10 labels utilize the same six reporter ions ranging from 126 to 131 Da. The difference between TMT11 and TMT10 is the splitting of the 131 (last) channel into 131-N and 131-C. The analysis of data clustering shows that TMT10 and TMT11-labeled tumor profiles are relatively mixed with each other, which allowed us to analyze proteomic profiles obtained using these two labels as a single dataset. However, we observed very strong clustering of data by the model of mass spectrometer which was even stronger than clustering by the cancer type. The Orbitrap Fusion Lumos is a tribrid mass spectrometer that combines three mass analyzers: quadrupole technology, Orbitrap, and linear ion trap. The Q Exactive HF-X and Q Exactive Plus include quadrupole technology and Orbitrap mass spectrometry. However, there are some technical differences between them, e.g., the resolving power is up to 240 and 140 kFWHM for Exactive HF-X and Q Exactive Plus, respectively. We demonstrated that the datasets produced by the Orbitrap Fusion Lumos, Q Exactive HF-X, and Q Exactive Plus have a different number of significant potential biomarkers (Orbitrap Fusion Lumos platform gave a ~2-fold higher proportion of potential proteomic biomarkers than the Q Exactive Plus engine, Table 4). Currently, we do not know whether this difference is related to platform-specific data quality or to the biological properties of the tissues investigated with the respective platforms. For the same reasons, we cannot correctly compare the potential biomarker capacities of the TMT10 and TMT11 labels. However, we believe that it has to be investigated in detail in the future to enable high-quality comparative combinatorial studies of proteomic datasets.

Furthermore, the resolution of the proteomic platforms investigated here in terms of the number of items for which expression can be quantitatively assessed is ~3.6-fold lower than for the transcriptomic data obtained by RNA sequencing [15]. However, the percentage of potential cancer type-specific biomarkers was comparable between proteomic (21–58%, average 39%) and transcriptomic (7–53%, average 26%) data at the level of single gene products (Table 2 and Table 3). Similarly, the percentage of proteomic pathway-based biomarkers was also similar to the transcriptomic results: 22–66% (average 44%) and 8–65% (average 33%), respectively (Table 2 and Table 3).

However, the proportion of potential survival biomarkers was dramatically lower for the proteomic data, where statistically significant potential biomarkers were found only in four of eight cancers (50%) versus 13 of 21 (62%) for the transcriptomic data, and their percentage was only up to 2.3% versus 36.3% in transcriptomic data (Figure 6 and Figure 7). For example, only six individual proteins and no molecular pathways were associated with overall survival in lung squamous cell carcinoma while no individual genes, 17 gene-centric pathways, and 154 classical pathways were associated at the transcriptomic level. At the same time, we could find survival biomarkers of pancreatic cancer only at the proteomic level (Figure 6 and Figure 7).

We used the same statistical criteria for both transcriptomic and proteomic data. However, despite the similar tumor stage distributions, the CPTAC and TCGA cohorts may differ significantly by treatment. The therapy used is not completely described, and standard treatment protocols may be not the same because the time gap between sample collections is about 10 years. This factor may impact survival analysis results.

This study used protein abundance data that correspond to the gene level. However, each gene may have multiple proteoforms due to alternative splicing and posttranslational modifications (PTMs). The presence of various proteoforms can have a significant impact on the potential use of a protein as a biomarker. To assess data complexity, we tested the kidney cancer phosphoproteomic CPTAC dataset PDC000128 using the COPF approach [64]. COPF is a data-driven method that detects groups of highly correlated peptides in bottom-up proteomic datasets. Such groups can, but do not have to represent unique, specific proteoforms. We found that 485 out of 4689 proteins (10.3%) have highly correlated groups of phosphopeptides (p-adjusted < 0.1). Moreover, to assess potential proteoforms, we need information about other PTMs for the same samples, that can substantially increase the number of proteins with highly correlated groups of peptides. Furthermore, methods for the detection of proteoforms in bottom-up proteomics should be developed and validated for different PTMs. Certainly, an analytical approach for bottom-up proteomics can be used to assess potential proteoform groups, however, top-down data are needed to detect specific proteoforms. We believe that with further accumulation of data on posttranslational modifications for a larger number of samples and cancer types, our biomarker assay should also be repeated at the level of different proteoforms.

In our study, the gene-centric pathways could identify cancer types better than their corresponding central genes (Figure 3). For some cancer types, they also provided a larger proportion of potential biomarkers than classical pathways, yet no clear overall trend could be identified.

On the other hand, in the case of potential survival biomarkers (Figure 9B,D and Figure 10B,D,F) pathways of either type did not show a high advantage over single genes (Figure 8). In terms of the percentage of successful potential biomarkers, single genes were the best category in six cancer types, whereas gene-centric and classical pathways were each on the top in eight cancer types.

We also speculate here that our approach can be employed not only to screen for cancer type or survival biomarkers but also to identify new therapeutic response biomarkers or tumorigenesis-associated gene networks. Overall, we found that the percentage of high-quality potential biomarkers was statistically significantly higher among the molecular pathways, both gene-centric and classical, than in individual genes. In turn, both types of pathways showed comparable performance. Thus, we conclude that pathway activation level is the advanced type of new generation of cancer biomarkers.

The potential biomarkers identified here may be of interest for molecular cancer research. By analyzing pathway activities, we can gain deeper insights into the pathophysiology of specific cancer types and unravel complex molecular networks that drive tumorigenesis. Moreover, the identification of new algorithmically constructed pathways with clinical relevance may enhance the search for novel drug targets and the development of more effective therapeutic interventions.

Furthermore, we believe that such momentary algorithmic computer building of pathways is a new credible alternative to time-consuming hypothesis-driven manual reconstruction of pathways and can replace it in the nearest future.

## 5. Limitations

Cancer type-specific differential expression of genes and their association with survival has been already extensively investigated in a number of previous reports. The primary focus of this study was to compare the biomarker predictive capacity of algorithmically constructed pathways with those of the previously established types of biomarkers: manually curated molecular pathways and single genes. We demonstrate here that both types of pathways significantly outperform single gene expression levels as potential biomarkers. However, prior to considering any clinical use of such putative biomarkers identified in our bioinformatic assay, they need to undergo further clinical validation on independent patient cohorts.

In this study, we used only uniformly generated proteomic datasets obtained during the CPTAC project, with sample-independent default normalization (subtraction of the median). To eliminate possible effects of different experimental references (the pooled samples), we used relative ratios of protein expression levels in tumor tissues to normal tissues. Unfortunately, we were unable to use correction for batch effects because in this case, the biological factor (cancer type) coincides with the batch factor. A single large proteomic dataset with different cancer types is required to assess differences between cancers without a potential batch factor.

## Figures and Tables

**Figure 1 proteomes-11-00026-f001:**
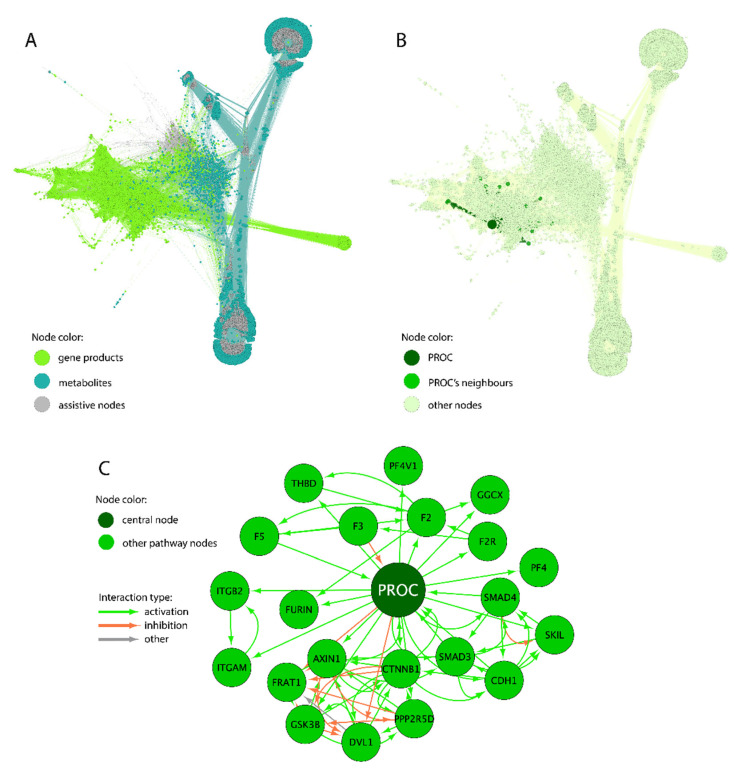
The human interactome model and example of gene-centric molecular pathway. (**A**) graph of the human whole interactome. The edges inherit node colors. Assistive nodes denote biochemical reactions and molecular transport if such processes include more than two components and cannot be visualized as an arrow between two participants. (**B**) example of a gene-centric pathway PROC. Projections of the central node (*PROC* gene product, dark green) and of the pathway members (green) are shown on the whole interactome. The rest of the interactome graph is shown as background. (**C**) Isolated view of PROC-centric pathway and its interacting components.

**Figure 2 proteomes-11-00026-f002:**
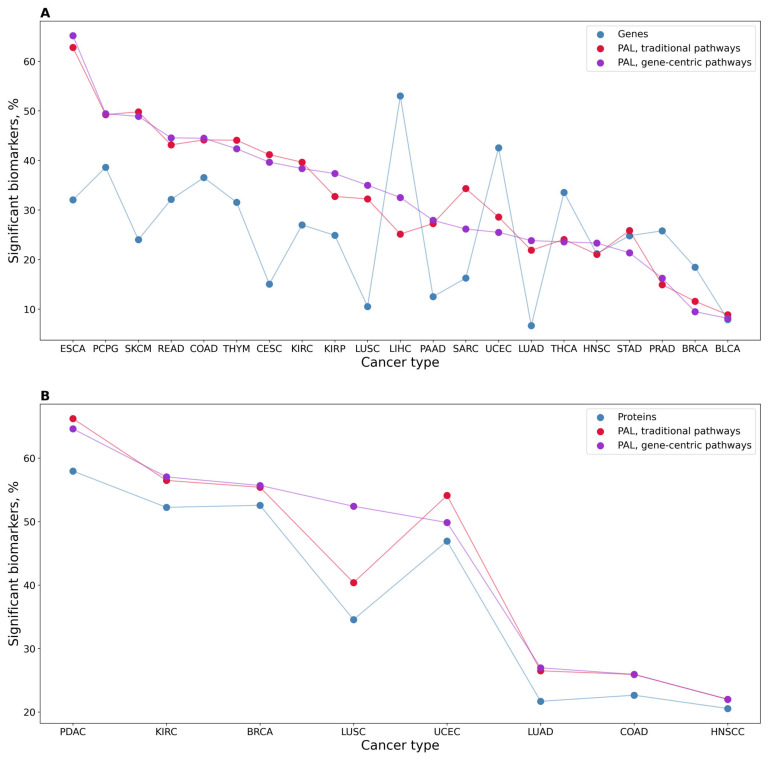
Percentage of significant potential cancer-type-specific biomarkers among the total number of tested items. (**A**) In transcriptomic datasets, 21 cancer types, 7470 gene-centric molecular pathways, 3022 classical molecular pathways, and 24,862 individual genes were investigated. (**B**) Protein expression analysis: eight cancer types, 7418 gene-centric molecular pathways, 2994 classical molecular pathways, and 6862 proteins.

**Figure 3 proteomes-11-00026-f003:**
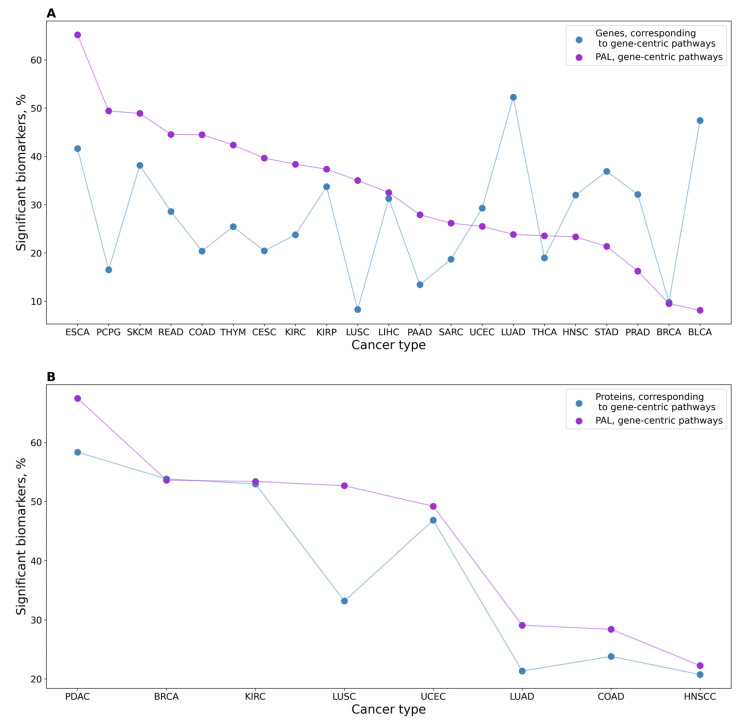
(**A**) Profile of percentage share of significant potential biomarkers for 21 cancer types under analysis for PALs of gene-centric pathways and expression levels of genes serving as their central nodes, using RNAseq data. (**B**) Profile of percentage share of significant potential biomarkers for eight cancer types for proteomic-based PALs of gene-centric pathways and expression levels of proteins serving as their central nodes.

**Figure 4 proteomes-11-00026-f004:**
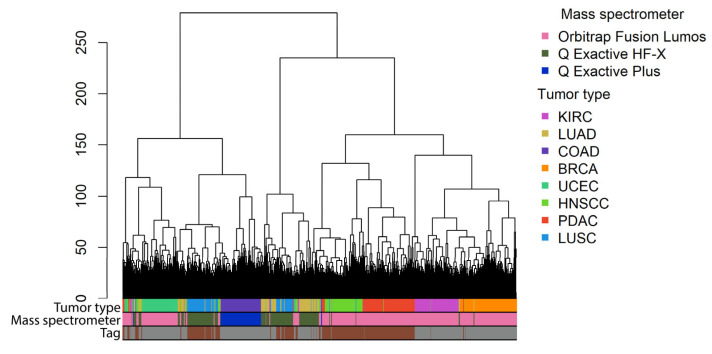
Clustering of log-transformed ratios of tumor profiles to average normal profiles. The height of dendrogram branches corresponds to the Euclidian distance between clusters. The clustering method is ward.d2.

**Figure 5 proteomes-11-00026-f005:**
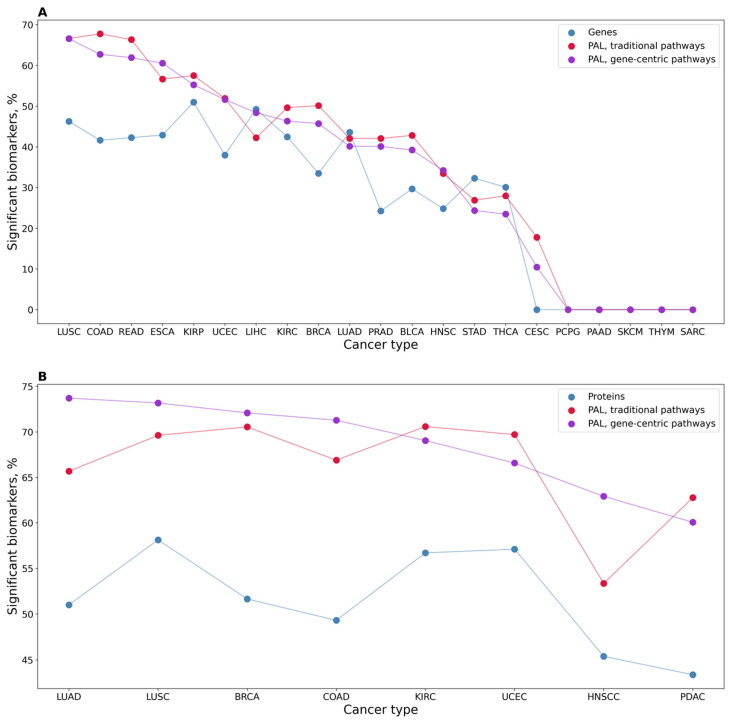
The percentage of significant potential tumor biomarkers out of the total number of tested items is shown. (**A**) The study investigates a total of 21 types of cancers, 7470 gene-centric molecular pathways, 3022 classical molecular pathways, and 24,862 individual genes present in transcriptomic datasets. (**B**) Protein expression analysis was performed for eight cancer types, the number of items tested for each type is available in Appendix A.

**Figure 6 proteomes-11-00026-f006:**
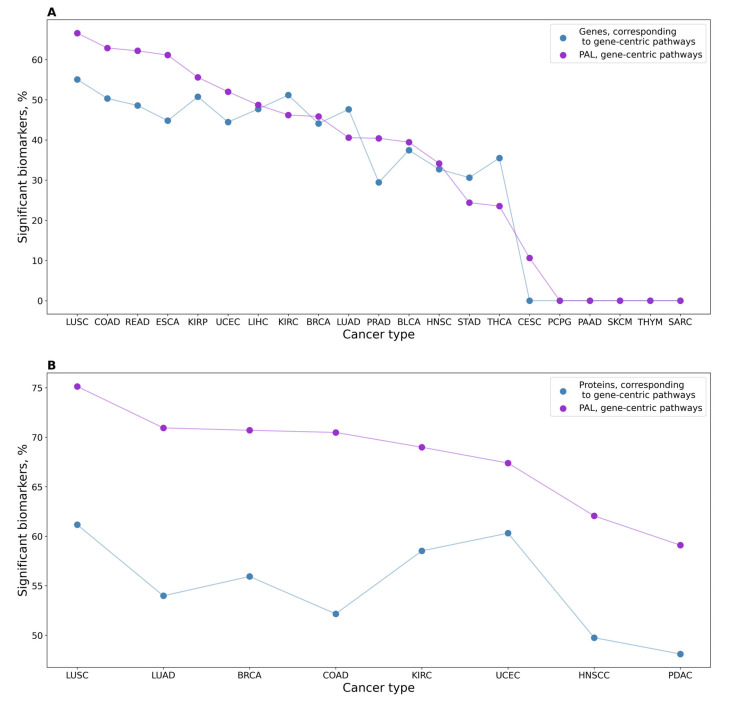
(**A**) Profile of the percentage share of significant potential tumor biomarkers for 21 cancer types analyzed for PALs of gene-centric pathways and expression levels of genes serving as their central nodes, using RNAseq data. (**B**) Profile of the percentage share of significant potential tumor biomarkers for eight cancer types for proteomic-based PALs of gene-centric pathways and expression levels of proteins serving as their central nodes.

**Figure 7 proteomes-11-00026-f007:**
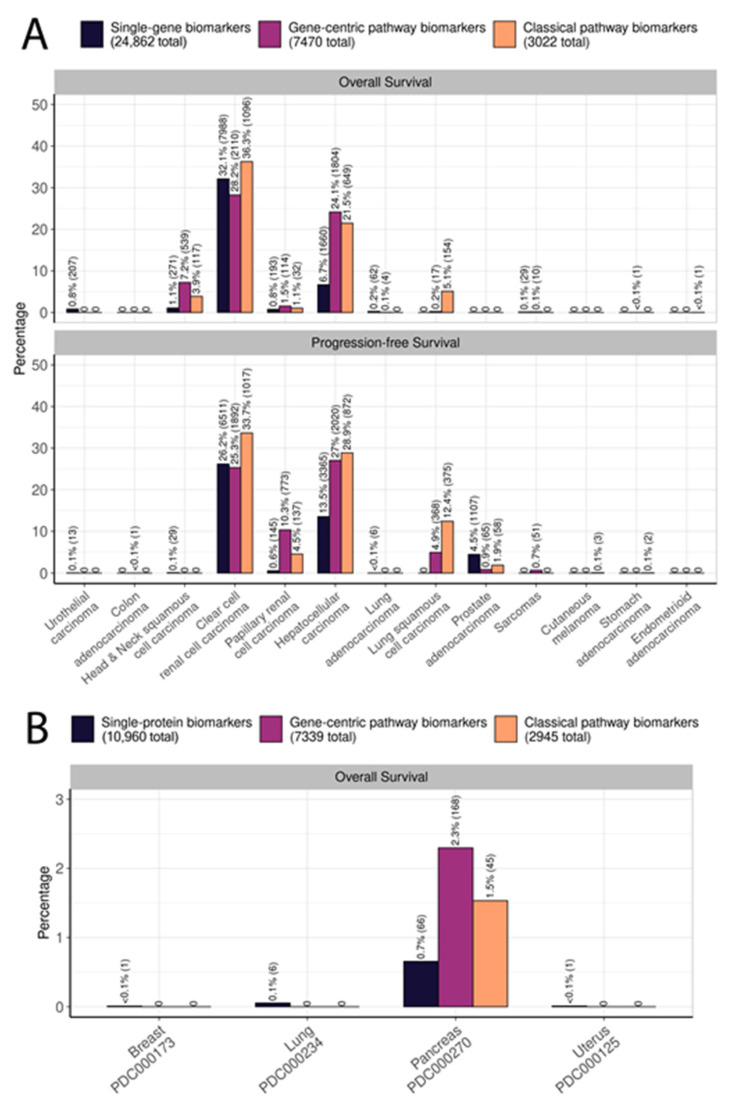
Percentage of potential prognostic biomarkers characterized by category, type of survival data, and cancer type. (**A**) Numbers of potential biomarkers found in the three biomarker categories by cancer type and by type of survival data using RNAseq data. (**B**) Numbers of potential biomarkers found in the three biomarker categories by cancer type and by type of survival data using proteomic data. Percentages refer to the fractions of statistically significant biomarkers within each category. Cancer types with no biomarkers detected are not shown.

**Figure 8 proteomes-11-00026-f008:**
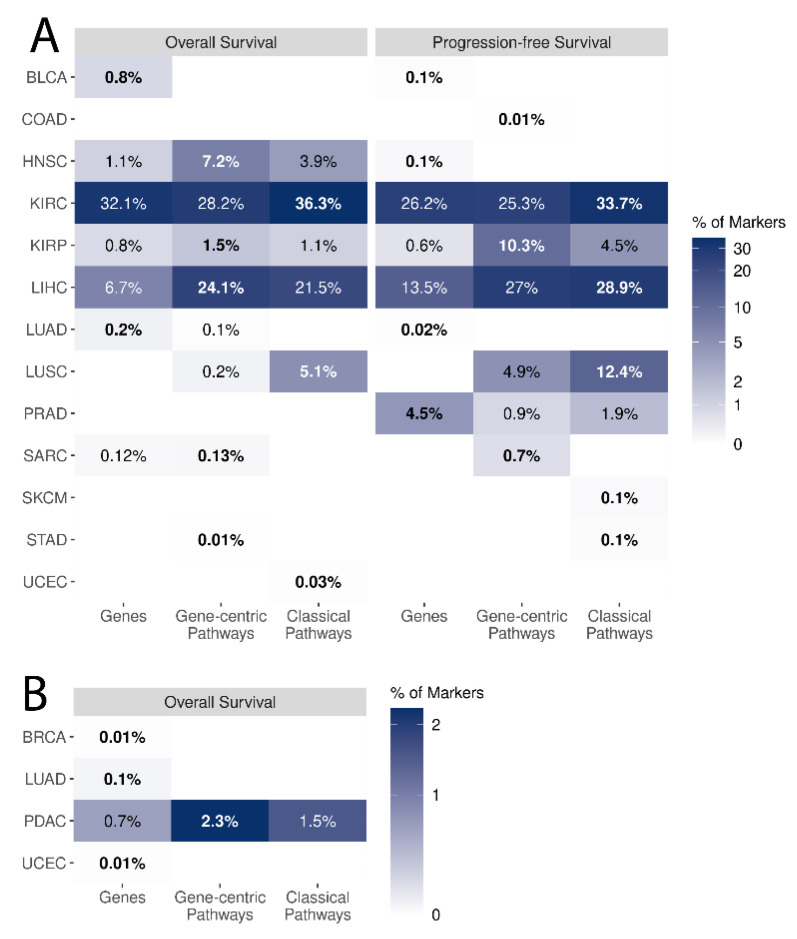
Percentage of potential prognostic biomarkers characterized by category, type of survival data, and cancer type. Percentage values are reflected by the logarithmic color scale and text labels. For zero values, the labels are not shown. In each cancer type, the percentage for the category with the highest percentage of potential biomarkers (winning position) is highlighted in bold. Cancer types with no biomarkers detected are not shown. Cancer-type abbreviations are given according to Table 1.

**Figure 9 proteomes-11-00026-f009:**
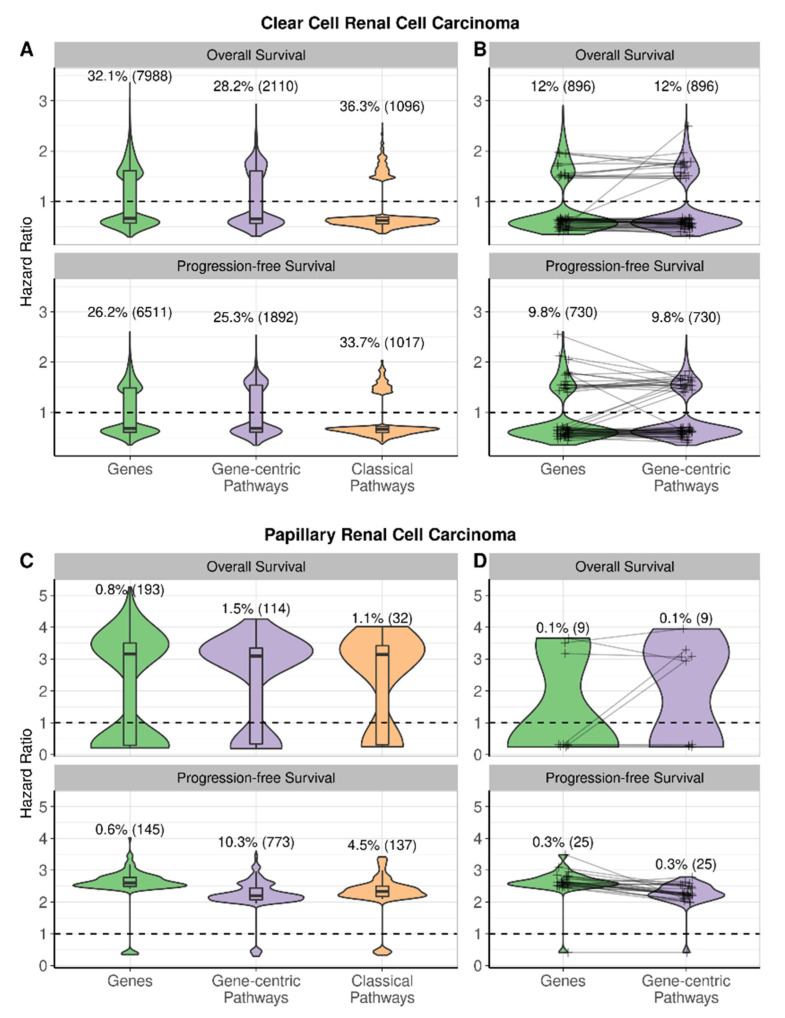
Distributions of hazard ratios of potential biomarkers in renal cell carcinomas by biomarker category and survival type. n is the number of potential biomarkers in the respective distribution. The dashed line at HR = 1 corresponds to no difference in survival. (**A**) HRs of genes, gene-centric pathways, and classical pathways significantly associated with overall and progression-free survival in clear cell renal cell carcinoma. (**B**) HRs of gene-centric pathways and their central genes, paired together, in cases where both were significantly associated with overall and progression-free survival in clear cell renal cell carcinoma. (**C**) HRs of genes, gene-centric pathways, and classical pathways significantly associated with overall and progression-free survival in papillary renal cell carcinoma. (**D**) HRs of gene-centric pathways and their central genes, paired together, in cases where both were significantly associated with overall and progression-free survival in papillary renal cell carcinoma. Panels (**A**,**C**) show all genes, gene-centric pathways, and classical pathways significantly associated with survival. Panels (**B**,**D**) show genes serving as central nodes in gene-centric pathways, connected by lines with their respective pathways, in cases where both were significantly associated with survival. In cases with more than 100 genes and pathways, only 100 randomly selected gene-pathway pairs are connected by lines.

**Figure 10 proteomes-11-00026-f010:**
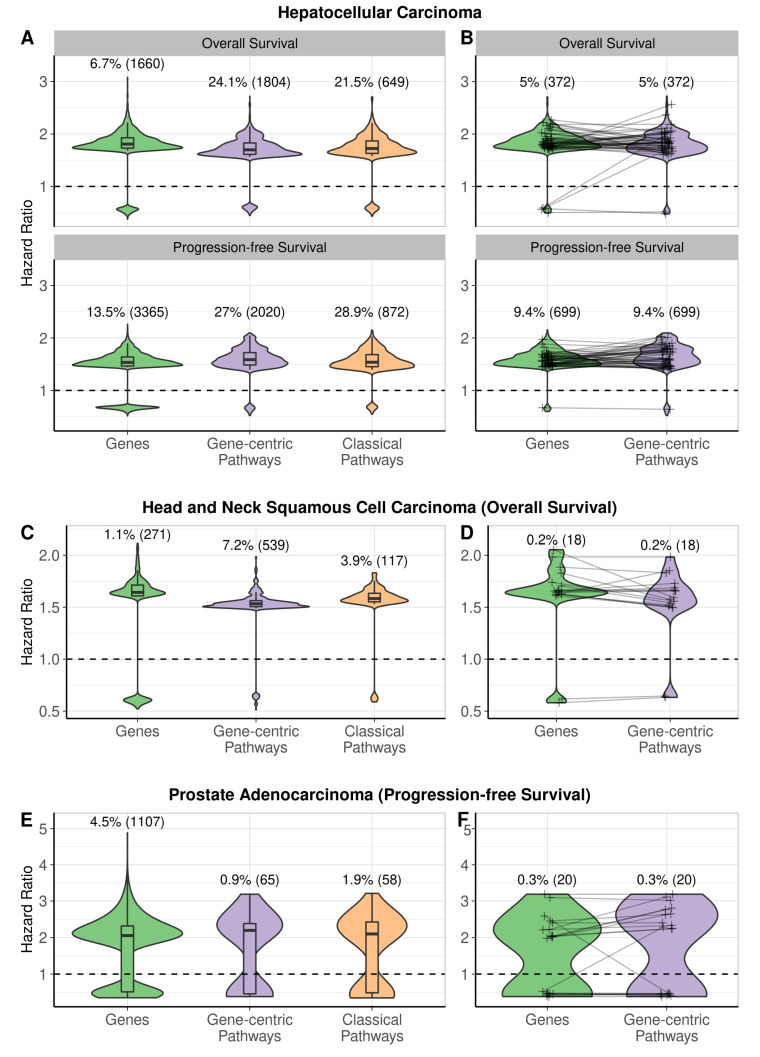
Distributions of hazard ratios of potential biomarkers in hepatocellular carcinoma, head and neck squamous cell carcinoma, and prostate adenocarcinoma by biomarker category and survival type. (**A**) HRs of genes, gene-centric pathways, and classical pathways significantly associated with overall and progression-free survival in hepatocellular carcinoma. (**B**) HRs of gene-centric pathways and their central genes, paired together, in cases where both were significantly associated with overall and progression-free survival in hepatocellular carcinoma. (**C**) HRs of genes, gene-centric pathways, and classical pathways significantly associated with overall and progression-free survival in head and neck squamous cell carcinoma. (**D**) HRs of gene-centric pathways and their central genes, paired together, in cases where both were significantly associated with overall and progression-free survival in head and neck squamous cell carcinoma. (**E**) HRs of genes, gene-centric pathways, and classical pathways significantly associated with overall and progression-free survival in prostate adenocarcinoma. (**F**) HRs of gene-centric pathways and their central genes, paired together, in cases where both were significantly associated with overall and progression-free survival in prostate adenocarcinoma. For the latter two cancer types, only overall survival and progression-free survival potential biomarkers are shown, respectively, as these were the only survival categories that had enough potential biomarkers to visualize a comparison; *n* is the number of potential biomarkers in the respective distribution. The dashed line at HR = 1 corresponds to no difference in survival. The panels (**A**,**C**,**E**) show all genes, gene-centric pathways, and classical pathways significantly associated with survival. The panels (**B**,**D**,**F**) show only genes serving as central nodes in gene-centric pathways, connected by lines with their respective pathways, in cases where both were significantly associated with survival. In cases with more than 100 genes and pathways, only 100 random gene-pathway pairs are connected by lines.

**Table 1 proteomes-11-00026-t001:** Overview of clinically annotated RNA sequencing and proteomic profiles used in this study. Projects marked with asterisks (*) were simplified by histological type for survival analysis: a dominant tumor type was identified, and samples of other histological types were excluded from the analysis.

Cancer Type	Abbreviation	RNAseq Data			Proteomic Data		
		Number of Tumor Samples	Number of Samples with Survival Data	Number of Tumor-Matched Normal Samples	Number of Tumor Samples	Number of Samples with Survival Data	Number of Normal Samples
Urothelial carcinoma	BLCA *	412	403	19	-	-	-
Infiltrating ductal carcinoma of the breast	BRCA *	1106	777	113	240	102	21
Cervical squamous cell carcinoma	CESC *	304	252	3	-	-	-
Colon adenocarcinoma	COAD *	478	452	41	95	-	100
Esophageal carcinoma	ESCA	162	162	11	-	-	-
Head and neck squamous cell carcinoma	HNSC	502	502	44	110	-	68
Clear cell renal cell carcinoma	KIRC	540	532	72	110	101	84
Papillary renal cell carcinoma	KIRP	290	290	32	-	-	-
Hepatocellular carcinoma	LIHC *	371	355	50	-	-	-
Lung adenocarcinoma	LUAD	537	516	59	113	105	102
Lung squamous cell carcinoma	LUSC	502	501	49	110	107	102
Infiltrating ductal adenocarcinoma of the pancreas	PAAD *	178	143	4	137	108	74
Pheochromocytoma	PCPG *	179	149	3	-	-	-
Prostate adenocarcinoma	PRAD *	500	484	52	-	-	-
Rectal adenocarcinoma	READ	166	165	10	-	-	-
Sarcomas	SARC	259	259	2	-	-	-
Cutaneous melanoma	SKCM	103	103	1	-	-	-
Stomach adenocarcinoma	STAD	375	375	32	-	-	-
Thyroid carcinoma	THCA *	504	496	59	-	-	-
Thymomas	THYM *	120	109	2	-	-	-
Endometrioid adenocarcinoma	UCEC *	553	401	35	103	88	30
**All cancer types**	**Total**	**8141**	**7426**	**693**	**1018**	**611**	**581**

**Table 2 proteomes-11-00026-t002:** Number of potential cancer-type-specific biomarkers for 21 cancer types under analysis.

TCGA Cancer ID	Marker Genes	Marker Classical Pathways	Marker Gene-Centric Pathways	TCGA Cancer ID	Marker Genes	Marker Classical Pathways	Marker Gene-Centric Pathways
**PCPG**	9601 (39%)	1488 (49%)	3691 (49%)	**LUSC**	2610 (10%)	974 (32%)	2614 (35%)
**BLCA**	1959 (8%)	268 (9%)	608 (8%)	**PAAD**	3108 (13%)	824 (27%)	2085 (28%)
**BRCA**	4590 (18%)	350 (12%)	709 (9%)	**PRAD**	6412 (26%)	451 (15%)	1212 (16%)
**CESC**	3743 (15%)	1244 (41%)	2961 (40%)	**SKCM**	5969 (24%)	1505 (50%)	3652 (49%)
**UCEC**	10,579 (43%)	864 (29%)	1904 (25%)	**STAD**	6168 (25%)	781 (26%)	1595 (21%)
**COAD**	9082 (37%)	1334 (44%)	3322 (44%)	**THYM**	7846 (32%)	1332 (44%)	3164 (42%)
**HNSC**	5263 (21%)	636 (21%)	2961 (40%)	**THCA**	8343 (34%)	727 (24%)	1760 (24%)
**KIRC**	6709 (27%)	1198 (40%)	2866 (38%)	**SARC**	4042 (16%)	1037 (34%)	1954 (26%)
**KIRP**	6186 (25%)	989 (33%)	2790 (37%)	**ESCA**	7963 (32%)	1898 (63%)	4869 (65%)
**LIHC**	13,180 (53%)	760 (25%)	2429 (33%)	**READ**	7988 (32%)	1304 (43%)	3328 (45%)
**LUAD**	1660 (7%)	661 (22%)	1780 (24%)	**Total**	**24,349**	**3020**	**7441**

**Table 3 proteomes-11-00026-t003:** Number of potential expression biomarkers for eight cancer types identified at the proteomic level.

CPTAC Project ID	Proteins	Classical Pathways	Gene-Centric Pathways	Label, TMT10/TMT11	Mass Spectrometer
**KIRC PDC000127**	3585 (52%)	4231 (57%)	1691 (56%)	TMT10	Orbitrap Fusion Lumos
**LUAD PDC000153**	1488 (22%)	2000 (27%)	793 (26%)	TMT10	Q Exactive HF-X
**COAD PDC000116**	1554 (23%)	1923 (26%)	776 (26%)	TMT10	Q Exactive Plus
**BRCA PDC000120**	3607 (53%)	4131 (56%)	1659 (55%)	TMT10	Orbitrap Fusion Lumos
**UCEC PDC000125**	3220 (47%)	3698 (50%)	1620 (54%)	TMT10	Orbitrap Fusion Lumos
**HNSC PDC000221**	1410 (21%)	1631 (22%)	659 (22%)	TMT11	Orbitrap Fusion Lumos
**LUSC PDC000234**	2371 (35%)	3888 (52%)	1209 (40%)	TMT11	Q Exactive HF-X
**PDAC PDC000270**	3977 (58%)	4793 (65%)	1983 (66%)	TMT11	Orbitrap Fusion Lumos
**Total**	**6742**	**2950**	**7343**		

**Table 4 proteomes-11-00026-t004:** Average percentage share of statistically significant proteomic expression biomarkers for datasets produced by three types of mass spectrometers.

Mass Spectrometer	Individual Proteins (%)	Classical Pathways (%)	Gene-Centric Pathways (%)
Orbitrap Fusion Lumos (5 datasets)	46	50	51
Q Exactive HF-X (2 datasets)	29	39	33
Q Exactive Plus (1 dataset)	22	27	26

**Table 5 proteomes-11-00026-t005:** The number of transcriptomic tumor potential biomarkers analyzed for 21 types of cancer.

TCGA Cancer ID	Marker Genes	Marker Classical Pathways	Marker Gene-Centric Pathways	TCGA Cancer ID	Marker Genes	Marker Classical Pathways	Marker Gene-Centric Pathways
**PCPG**	0 (0%)	0 (0%)	0 (0%)	**LUSC**	11,508 (46%)	2013 (67%)	4989 (67%)
**BLCA**	7379 (30%)	1294 (43%)	2939 (39%)	**PAAD**	0 (0%)	0 (0%)	0 (0%)
**BRCA**	8328 (33%)	1515 (50%)	3427 (46%)	**PRAD**	6031 (24%)	1271 (42%)	3005 (40%)
**CESC**	0 (0%)	537 (18%)	784 (10%)	**SKCM**	0 (0%)	0 (0%)	0 (0%)
**UCEC**	9445 (38%)	1568 (52%)	3868 (52%)	**STAD**	8032 (32%)	813 (27%)	1825 (24%)
**COAD**	10,351 (42%)	2048 (68%)	4701 (63%)	**THYM**	0 (0%)	0 (0%)	0 (0%)
**HNSC**	6172 (25%)	1011 (33%)	2563 (34%)	**THCA**	7481 (30%)	845 (28%)	1759 (23%)
**KIRC**	10,559 (42%)	1500 (50%)	3471 (46%)	**SARC**	0 (0%)	0 (0%)	0 (0%)
**KIRP**	12,673 (51%)	1738 (58%)	4140 (55%)	**ESCA**	10,671 (43%)	1712 (57%)	4537 (61%)
**LIHC**	12,245 (49%)	1276 (42%)	3627 (48%)	**READ**	10,510 (42%)	2005 (66%)	4640 (62%)
**LUAD**	10,836 (44%)	1271 (42%)	3008 (40%)	**Total**	**24,548**	**3021**	**7466**

**Table 6 proteomes-11-00026-t006:** Number of expression biomarkers for eight cancer types identified at the proteomic level.

CPTAC Project ID	Proteins	Classical Pathways	Gene-Centric Pathways	Label, TMT10/TMT11	Mass Spectrometer
**KIRC PDC000127**	5649 (57%)	2078 (71%)	5054 (69%)	TMT10	Orbitrap Fusion Lumos
**LUAD PDC000153**	5622 (51%)	1936 (66%)	5419 (74%)	TMT10	Q Exactive HF-X
**COAD PDC000116**	3657 (49%)	1927 (67%)	5116 (71%)	TMT10	Q Exactive Plus
**BRCA PDC000120**	5417 (52%)	2076 (71%)	5299 (72%)	TMT10	Orbitrap Fusion Lumos
**UCEC PDC000125**	6147 (57%)	2047 (70%)	4889 (67%)	TMT10	Orbitrap Fusion Lumos
**HNSC PDC000221**	4650 (45%)	1571 (53%)	4607 (63%)	TMT11	Orbitrap Fusion Lumos
**PDAC PDC000270**	4387 (43%)	1844 (63%)	4401 (60%)	TMT11	Orbitrap Fusion Lumos
**LUSC PDC000234**	6673 (58%)	2055 (70%)	5386 (73%)	TMT11	Q Exactive HF-X
**Total**					

**Table 7 proteomes-11-00026-t007:** Biomarkers found in different gene network approaches.

Reference	Disease	Input Data	Results	Gene Network Construction Method
[52]	Lung squamous cell carcinoma	RNA expression data for 15 patients	Seven out of 24 gene networks generated from differentially expressed genes were correlated with overall survival	http://www.ingenuity.com (accessed on 20 May 2023)
[53]	Gastric cancer	RNA expression data for 265 (TCGA) + 200 (GSE15459) patients	Gene correlation network of 249 genes significantly associated with overall survival. Four functional network components were highlighted	http://baderlab.org/Software/ExpressionCorrelation (accessed on 20 May 2023))
[54]	Colon cancer	RNA expression data for 461 patients (GSE39582)	11 gene networks associated with tumor grade and progression-free survival	WGCNA
[55]	Lung adenocarcinoma	RNA expression data for 82 patients	Gene network enriched with cell cycle-related genes correlated with tumor grade and overall survival	WGCNA
[56]	Renal clear cell carcinoma	RNA expression data for 533 patients (TCGA)	From 12 gene networks, two (“cell cycle” and “p53 signaling” pathways) associated with overall survival	WGCNA
[57]	Bladder cancer	RNA expression data for 414 patients (TCGA)	Protein interactions of 77 genes: 37 genes formed a network related to overall survival	WGCNA + STRING
This study; cancer type markers	Pan-cancer analysis: 21 cancer types	RNA expression data for 8141 patients (TCGA)	For 14 of 21 cancer types, both gene-centric and classical pathways were better cancer type-specific biomarkers than individual genes. In total, 3020 classical and 7441 genecentric pathways were identified as cancer type-specific biomarkers.	Classical and gene-centric molecular pathways
This study; cancer type markers	Pan-cancer analysis: 8 cancer types	Proteomic data for 1018 patients (CPTAC)	For all cancer types, both gene-centric and classical pathways had a higher percentage of significant biomarkers than single proteins. In total, 2950 classical and 7343 gene-centric pathways were identified as cancer-type-specific biomarkers.	Classical and gene-centric molecular pathways
This study; survival markers	Pan-cancer analysis: 21 cancer types	RNA expression data for 7426 patients (TCGA)	For overall survival, the highest percentage of biomarkers was observed in five and three cancer types for gene-centric and classical pathways, respectively. For progression-free survival, the advantage for gene-centric and classical pathways was shown for three and five cancer types, respectively.	Classical and gene-centric molecular pathways
This study; survival markers	Pan-cancer analysis: 6 cancer types	Proteomic data for 611 patients (CPTAC)	Statistically significant survival pathway-based biomarkers were found for pancreatic cancer (168 gene-centric and 45 classical pathways). Gene-centric pathways showed the highest percentage of biomarkers identified.	Classical and gene-centric molecular pathways

## Data Availability

The data presented in this study are available in this article in the Appendix A.

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
