# Peer review of "Algorithmically Reconstructed Molecular Pathways as the New Generation of Prognostic Molecular Biomarkers in Human Solid Cancers"

_proteomes, 2023, doi:10.3390/proteomes11030026_

Round 1

Reviewer 1 Report

Zolotovskaia and colleagues have submitted a research article entitled “Algorithmically Reconstructed Molecular Pathways as the New Generation of Prognostic Molecular Biomarkers in Human Solid Cancers” for publication in Proteomes. 

In the manuscript, the authors described an approach to constructing gene-centric molecular pathways, followed by assessing their value as cancer biomarkers. The results were compared with "classical" molecular pathways and single molecular features per se. The work combines interactome analysis, pathway activation levels, and previously published transcriptomics and proteomics data. Therefore, the manuscript is aligned with the journal's scope, highlighting a multi-/transdisciplinary approach based on different ‘omics’ techniques and computational science. Therefore, such a publication would interest Proteomes readers, but I feel additional work would be needed before the publication.  

Since I am not an expert in bioinformatics, I cannot comment on the methodology followed, though I would provide insights from the biomarker/omics perspective. My suggestions/ comments are detailed below.

1.      This is an exploratory study of a new methodology that can be used to define biomarkers. Although the authors suggest that the method leads to the identification of numerous biomarkers if such biomarkers would be of clinical applicability and what would be their performance and benefit over the standard of care remains to be determined. Therefore, this approach's actual value in delivering clinically relevant biomarkers must be confirmed in independent studies targeting specific contexts of use. A comment on that in the limitations section would be of value.

2.      The current study set-up, based on defining biomarkers by comparing specific cancer types with other available datasets (other cancer types and controls), is questionable and clinically not relevant. Why was such an approach followed instead of a case-control comparison between tumor vs. no-tumor (per cancer or in pooled analysis), which seems more reasonable? The justification should be provided in the text. 

3.      Numerous biomarkers are claimed to be defined by three methodologies compared in this study. But the authors mainly focus on assessing the number or percentage of biomarkers, which may, to some extent, reflect the statistical power, and variability. The manuscript led to the understanding that the higher the number of biomarkers detected, the better, which might be inaccurate. Other measures assessing the validity of biomarkers from three different approaches should be applied e.g., were these markers previously reported in the literature? 

4.      Although some biomarkers show prognostic value, such results require further validation in independent cohorts in a well-defined context; which should be acknowledged. 

5.      The comparison between biomarkers generated based on transcriptomics and proteomics data might be highly biased by the disproportionate number of datasets and additional variability introduced by combining different methodological setups for proteomics study. The lower number of datasets in proteomics (plus additional variability) leads to a lower number of significant features (at the single molecule level), then lower pathway coverage, and likely a lower number of significantly affected pathways (either classical or gene-centric). The numbers for datasets with survival data are even smaller and may explain the observations.

Since authors report on the differences between proteomics datasets, special attention should be given to removing these effects by using normalisation / batch effect correction; or finding other suitable datasets from PRIDE or other proteomics repositories, which I encourage to do. Otherwise, the presented results might be misleading. 

6.      Authors should be more precise when indicating which underlying data are used to define biomarkers. 

7.      Network medicine is gaining attention, given the information available from different omics datasets and the underlying molecular complexity of disease-associated mechanisms. The proposed approach and developed interactome network (subnetworks) would also be of value in understanding disease pathophysiology. To use these data by the scientific community, I recommend that authors make available data on human interactome and defined gene-centric pathways (subnetworks).

8.      Authors should justify the selection of pathway databases to construct the interactome.

9.      A comment on how those data can also be exploited (to understand pathophysiology, define new drug targets etc.) could be added to the discussion to broaden the perspective.

10.  It is known by now that combination of biomarkers (biomarker signatures) outperforms single biomarkers. So, not unexpectedly, when using numerous features within a pathway, the performance is expected to be higher (than for single molecular features). A comment on that aspect would be beneficial. 

11.  Table 1 presents an overview of the number of samples for which proteomics profiles are available. Statistical analysis is not given. Please modify the legend e.g. “Overview of clinically annotated RNA sequencing and proteomic profiles used in this study”. In addition to the text, please explain the meaning of “*” (abbreviation column) in the legend or table footnote.

12.  Figure 1: Please explain “assistive nodes” in the legend and provide examples of “other” interaction type.

Minor editing of English language required.

Author Response

Zolotovskaia and colleagues have submitted a research article entitled “Algorithmically Reconstructed Molecular Pathways as the New Generation of Prognostic Molecular Biomarkers in Human Solid Cancers” for publication in Proteomes. 

In the manuscript, the authors described an approach to constructing gene-centric molecular pathways, followed by assessing their value as cancer biomarkers. The results were compared with "classical" molecular pathways and single molecular features per se. The work combines interactome analysis, pathway activation levels, and previously published transcriptomics and proteomics data. Therefore, the manuscript is aligned with the journal's scope, highlighting a multi-/transdisciplinary approach based on different ‘omics’ techniques and computational science. Therefore, such a publication would interest Proteomes readers, but I feel additional work would be needed before the publication.  

Since I am not an expert in bioinformatics, I cannot comment on the methodology followed, though I would provide insights from the biomarker/omics perspective. My suggestions/ comments are detailed below.

  1. This is an exploratory study of a new methodology that can be used to define biomarkers. Although the authors suggest that the method leads to the identification of numerous biomarkers if such biomarkers would be of clinical applicability and what would be their performance and benefit over the standard of care remains to be determined. Therefore, this approach's actual value in delivering clinically relevant biomarkers must be confirmed in independent studies targeting specific contexts of use. A comment on that in the limitations section would be of value.

-We thank the Reviewer for his/her thorough analysis and useful recommendation. We agree and in the future a fraction of the biomarkers identified can be validated in each cancer type in a series of separate experiments. However, our current focus is comparison of an overall predictive capacity of algorithmically constructed molecular pathways, manually curated pathways, and single genes.  We added the corresponding comment in the Limitations section:

Cancer type-specific differential expression of genes and their association with survival has been already extensively investigated in a number of previous reports. The primary focus of this study was to compare biomarker predictive capacity of algorithmically constructed pathways with those of the previously established types of biomarkers: manually curated molecular pathways, and single genes. We demonstrate here that both types of pathways significantly outperform single gene expression levels as the potential biomarkers. However, prior to considering any clinical use of such putative biomarkers identified in our bioinformatic assay, they need to undergo further clinical validation on independent patient cohorts.

  1. The current study set-up, based on defining biomarkers by comparing specific cancer types with other available datasets (other cancer types and controls), is questionable and clinically not relevant. Why was such an approach followed instead of a case-control comparison between tumor vs. no-tumor (per cancer or in pooled analysis), which seems more reasonable? The justification should be provided in the text. 

-We thank the Reviewer for this suggestion. We now added the comparison between normal and tumor samples as the specific subsection of the Results. However, we prefer to leave the comparison of cancer types in the text because cancer type-specific changes may be important for better understanding of tissue-specific aspects of carcinogenesis. The explanation was added to the Discussion:

“The cancer type-specific biomarkers may be important for better understanding of tissue-specific aspects of carcinogenesis.”

  1. Numerous biomarkers are claimed to be defined by three methodologies compared in this study. But the authors mainly focus on assessing the number or percentage of biomarkers, which may, to some extent, reflect the statistical power, and variability. The manuscript led to the understanding that the higher the number of biomarkers detected, the better, which might be inaccurate. Other measures assessing the validity of biomarkers from three different approaches should be applied e.g., were these markers previously reported in the literature? 

-Indeed, gene expression biomarkers for cancer type(s) and/or survival were extensively published in the literature. However, no such analysis has been reported for the molecular pathway activation levels; therefore, we cannot use as the gene/pathway biomarker type comparison criterion such previous mentions in the literature.

However, we now added new statistical metrics to assess quality of the biomarkers: comparison of AUC and p-values for all pathways and genes tested. We found that medians of distributions showed somewhat better values for the pathway-based biomarkers.

  1. Although some biomarkers show prognostic value, such results require further validation in independent cohorts in a well-defined context; which should be acknowledged. 

-(see also our reply on point (1)) We agree. We now added the following to the manuscript:

Cancer type-specific differential expression of genes and their association with survival has been already extensively investigated in a number of previous reports. The primary focus of this study was to compare biomarker predictive capacity of algorithmically constructed pathways with those of the previously established types of biomarkers: manually curated molecular pathways, and single genes. We demonstrate here that both types of pathways significantly outperform single gene expression levels as the potential biomarkers. However, prior to considering any clinical use of such putative biomarkers identified in our bioinformatic assay, they need to undergo further clinical validation on independent patient cohorts.

  1. The comparison between biomarkers generated based on transcriptomics and proteomics data might be highly biased by the disproportionate number of datasets and additional variability introduced by combining different methodological setups for proteomics study. The lower number of datasets in proteomics (plus additional variability) leads to a lower number of significant features (at the single molecule level), then lower pathway coverage, and likely a lower number of significantly affected pathways (either classical or gene-centric). The numbers for datasets with survival data are even smaller and may explain the observations.

Since authors report on the differences between proteomics datasets, special attention should be given to removing these effects by using normalisation / batch effect correction; or finding other suitable datasets from PRIDE or other proteomics repositories, which I encourage to do. Otherwise, the presented results might be misleading. 

-Indeed, this is an important point. In this study we used only uniformly generated datasets obtained during CPTAC project, with sample-independent default normalization (subtraction of median). To eliminate possible effects of different experimental references (the pooled samples), we used relative ratios of protein expression levels in tumor tissue to normal tissues. Unfortunately, we were unable to use correction for batch effects because in this case the biological factor (cancer type) coincides with the batch factor. And unfortunately we are not aware of any available alternative uniformly obtained single large proteomic dataset with different cancer types.

  1. Authors should be more precise when indicating which underlying data are used to define biomarkers. 

-Thank you. We now checked the text and specified the data used for each analysis.

  1. Network medicine is gaining attention, given the information available from different omics datasets and the underlying molecular complexity of disease-associated mechanisms. The proposed approach and developed interactome network (subnetworks) would also be of value in understanding disease pathophysiology. To use these data by the scientific community, I recommend that authors make available data on human interactome and defined gene-centric pathways (subnetworks).

-We agree. The human interactome model we used in this study is publicly available following reference [31] for our previously published paper where we assessed biomarker capacity of gene-centric pathways for human gliomas. Additionally, we now added the reconstructed gene-centric pathways to the public pathway databank available through the link open.oncobox.com.

  1. Authors should justify the selection of pathway databases to construct the interactome.

 -We used the biggest available collection of molecular interaction-validated molecular pathways previously described in [30].This has been now added to the text.

  1. A comment on how those data can also be exploited (to understand pathophysiology, define new drug targets etc.) could be added to the discussion to broaden the perspective.

-We are grateful for this advice. We now added the following to Discussion:

“The biomarkers identified here may be of interest for molecular cancer research. By analyzing pathway activities, we can gain deeper insights into the pathophysiology of specific cancer types and unravel complex molecular networks that drive tumorigenesis. Moreover, identification of new algorithmically constructed pathways with clinical relevance may enhance finding novel drug targets and developing more effective therapeutic interventions.”

  1. It is known by now that combination of biomarkers (biomarker signatures) outperforms single biomarkers. So, not unexpectedly, when using numerous features within a pathway, the performance is expected to be higher (than for single molecular features). A comment on that aspect would be beneficial. 

-We are grateful for this comment. Indeed, cumulative metrics like pathway activation level or gene signatures act more robust as the biomarkers and have a greater performance compared to single gene or protein expression levels. This trend was demonstrated in many previous studies, many of which were cited in the Introduction. Also, we added the corresponding comment to Discussion:

“It is now widely accepted that combination of biomarkers, such as gene signatures or pathways, is more robust and performs better than using individual genes or proteins. Our results confirm this trend. “

  1. Table 1 presents an overview of the number of samples for which proteomics profiles are available. Statistical analysis is not given. Please modify the legend e.g. “Overview of clinically annotated RNA sequencing and proteomic profiles used in this study”.In addition to the text, please explain the meaning of “*” (abbreviation column) in the legend or table footnote.

-We modified the table description accordingly and added the explanation of the meaning of asterisks:

“Projects marked with asterisks (*) were simplified by histological type for survival analysis: a dominant tumor type was identified, and samples of other histological types were excluded from the analysis.”

  1. Figure 1: Please explain “assistive nodes” in the legend and provide examples of “other” interaction type.

-Thank you. The following explanation has been added:

“Assistive nodes denote biochemical reactions and molecular transport if such processes include more than two components and cannot be visualized as an arrow between two participants. “Other” interaction types include interactions which cannot be related to activation or inhibition. Examples of “other” interaction type are “SubPathway Interaction”, “Complex Assembly”, “Compound”, “Indirect values”.”

Reviewer 2 Report

In this paper, the authors used a human interactome model involving 7,470 human gene products to algorithmically reconstruct molecular pathways, termed gene-centric pathways, centered around each gene. They evaluated their general biomarker characteristics compared with the previous generation of 3,022 molecular pathways with the transcripts of 24,862 individual genes. They investigated biomarker associations with tumor type, overall, and progression-free survival in 21 human cancer types using RNA sequencing and proteomic data for 8,141 and 1,117 samples, respectively. They found that both types of pathways showed comparable performance. However, the percentage of RNA biomarkers was statistically significantly higher for both types of pathways than for individual genes. In addition, while the percentage of cancer type-specific biomarkers was comparable between proteomic and transcriptomic levels, the proportion of survival biomarkers was dramatically lower for the proteomic data: up to only 2.3% versus as much as 36.3% in transcriptomic data. The authors conclude that pathway activation level is the advanced type of cancer biomarkers for RNA data. The momentary algorithmic computer building of pathways is a new credible alternative to time-consuming hypothesis-driven manual pathway reconstruction. The paper is well-written, and the findings are very interesting. I recommend the paper for publication with minor revisions described below.

1.       Change “ITRAQ” to  ”iTRAQ” in lines 163, 171.

2.       Table 3: It is unclear why there are two lines in the middle. To separate TMT10 from TMT11, one line should be removed.

3.       The font size is too small, from lines 264 to 282.

4.       Table 4: Move the numbers to the middle of each lane.

5.       Table 5: the bottom line that separates this table from the text is missing.

Author Response

Reviewer 2

Comments and Suggestions for Authors

In this paper, the authors used a human interactome model involving 7,470 human gene products to algorithmically reconstruct molecular pathways, termed gene-centric pathways, centered around each gene. They evaluated their general biomarker characteristics compared with the previous generation of 3,022 molecular pathways with the transcripts of 24,862 individual genes. They investigated biomarker associations with tumor type, overall, and progression-free survival in 21 human cancer types using RNA sequencing and proteomic data for 8,141 and 1,117 samples, respectively. They found that both types of pathways showed comparable performance. However, the percentage of RNA biomarkers was statistically significantly higher for both types of pathways than for individual genes. In addition, while the percentage of cancer type-specific biomarkers was comparable between proteomic and transcriptomic levels, the proportion of survival biomarkers was dramatically lower for the proteomic data: up to only 2.3% versus as much as 36.3% in transcriptomic data. The authors conclude that pathway activation level is the advanced type of cancer biomarkers for RNA data. The momentary algorithmic computer building of pathways is a new credible alternative to time-consuming hypothesis-driven manual pathway reconstruction. The paper is well-written, and the findings are very interesting. I recommend the paper for publication with minor revisions described below.

-We cordially thank the Reviewer for the thorough analysis and positive evaluation of our manuscript.  We did our best to improve the text according to the Reviewer’s recommendations.

  1. Change “ITRAQ” to ”iTRAQ” in lines 163, 171.

-Done.

  1. Table 3: It is unclear why there are two lines in the middle. To separate TMT10 from TMT11, one line should be removed.

-Done, additional lines removed.

  1. The font size is too small, from lines 264 to 282.

 -Done, font size increased.

  1. Table 4: Move the numbers to the middle of each lane.

-Done

  1. Table 5: the bottom line that separates this table from the text is missing.

-We updated the table style and added the bottom line.
